# Smoking is associated with increased risk of cardiovascular events, disease severity, and mortality among patients hospitalized for SARS-CoV-2 infections

Ram Poudel[1], Lori B. Daniels[2], Andrew P. DeFilippis[1,3], Naomi M. Hamburg[1,4], Yosef Khan[1], Rachel J. Keith[1,5], Revanthy Sampath Kumar[2], Andrew C. Strokes[1,4], Rose Marie Robertson[1], Aruni Bhatnagar[1,5]*

1 American Heart Association Tobacco Regulation Center, Dallas, TX, United States of America,
2 University of California, San Diego, CA, United States of America, 3 Vanderbilt University, Nashville, TN, United States of America, 4 Boston University, Boston, MA, United States of America, 5 University of Louisville, Louisville, KY, United States of America

* aruni@louisville.edu

**Data Availability Statement:** The data underlying the results presented in the study are available from the American Heart Association COVID-19

## Abstract

The clinical sequalae of SARS-CoV-2 infection are in part dependent upon age and pre-existing health conditions. Although the use of tobacco products decreases cardiorespiratory fitness while increasing susceptibility to microbial infections, limited information is available on how smoking affects COVID-19 severity. Therefore, we examined whether smokers hospitalized for COVID-19 are at a greater risk for developing severe complications than non-smokers. Data were from all hospitalized adults with SARS-CoV-2 infection from the American Heart Association's Get-With-The-Guidelines COVID-19 Registry, from January 2020 to March 2021, which is a hospital-based voluntary national registry initiated in 2019 with 122 participating hospitals across the United States. Patients who reported smoking at the time of admission were classified as smokers. Severe outcome was defined as either death or the use of mechanical ventilation. Of the 31,545 patients in the cohort, 6,717 patients were 1:2 propensity matched (for age, sex, race, medical history, medications, and time-frame of hospital admission) and classified as current smokers or non-smokers according to admission data. In multivariable analyses, after adjusting for sociodemographic characteristics, medical history, medication use, and the time of hospital admission, patients self-identified as current smokers had higher adjusted odds of death (adjusted odds ratio [aOR], 1.41; 95% CI, 1.21–1.64), the use of mechanical ventilation (aOR 1.15; 95% CI 1.01–1.32), and increased risk of major adverse cardiovascular events (aOR, 1.27; 95% CI 1.05–1.52). Independent of sociodemographic characteristics and medical history, smoking was associated with a higher risk of severe COVID-19, including death.

Registry (https://www.heart.org/en/professional/quality-improvement/covid-19-cvd-registry).

**Funding:** Work in the laboratory of authors (RP, APD, NHM, RJK, ACS, RMR, and AB) is supported by a grant from National Institutes of Health and the Center for Tobacco Products (HL120163). The funders had no role in study design, data collection, and analysis, decision to publish, or preparation of the manuscript.

**Competing interests:** The authors have declared that no competing interests exist.

## Introduction

Prior research has demonstrated that smoking and the use of other tobacco products is associated with cardiorespiratory injury [1, 2], which is characterized by endothelial dysfunction [3], autonomic dysregulation [4], alveolar injury, and decreased lung capacity [5]. Over time, these changes accumulate and lead to an increase in the risk of developing lung cancer, emphysema, chronic obstructive pulmonary disease (COPD), as well as cardiovascular disease. In comparison with non-smokers, smokers are more vulnerable to respiratory infections such as influenza [6]. Nevertheless, whether smoking exacerbates the adverse consequences of SARS-CoV-2 infection remains unclear.

In addition to pneumonia and acute respiratory distress syndrome, SARS-CoV-2 triggers extensive extra-pulmonary injury [7]. Individuals with pre-existing conditions such as hypertension and diabetes experience a higher frequency of COVID-19 respiratory and extra-pulmonary adverse outcomes than those with low cardiovascular disease risk or without diabetes [8], but the effects of smoking on COVID-19 severity remain poorly understood.

Previous assessments of the impact of smoking on COVID-19 severity have yielded uncertain results. Early data from COVID-19 patients from China identified current smoking as a risk factor for disease progression [9]. Subsequently, some studies reported that smoking was an independent predictor of mortality [10], and that tobacco use predicts mortality [11]. However, other studies have reported that smoking was not associated with COVID-19 [12] or that that current smoking rates among COVID-19 patients were below the general population [13]. A recent meta-analysis of 32 studies concluded that the data for an association between current smoking and greater COVID severity or mortality were inconclusive, and favored no important associations with hospitalization and mortality [14]. Many methodological differences can account for such disparate observations including differences in sample size, the range of facilities examined, varying comparator groups, and not controlling for confounders such as age. Therefore, the purpose of the present study was to examine the effects of smoking history on the severity of COVID-19 among a large cohort of patients hospitalized for COVID-19 from a broad group of hospitals across the United States.

## Methods

### Data sources and collection (get with the guidelines cohort)

The data for this study were collected from the American Heart Association (AHA) COVID-19 Cardiovascular Disease (CVD) registry. Details of the AHA COVID-19 CVD Registry have been published previously [15]. The AHA COVID-19 CVD Registry was implemented in 2020 to gather data specific to all patients hospitalized with COVID-19 as part of the Get-With-The-Guidelines (GWTG) quality improvement program. This registry was provided free to all U.S. hospitals caring for adults with active COVID-19 and with the infrastructure to support accurate data collection. The GWTG program is a voluntary, in-hospital quality improvement initiative by the AHA. The data collection and coordination for GTWG program are managed by IQVIA (Parsippany, New Jersey). The AHA COVID-19 CVD registry collects more than 200 data elements of patients using case report forms (CRFs) [16]. (https://www.heart.org/en/professional/quality-improvement/covid-19-cvd-registry).

### Study period, population, and patient selection

We included all patients 18 years and older admitted to one of the 122 AHA COVID-19 CVD Registry participating hospitals from January 14, 2020 to March 31, 2021. Patients who left the hospital against medical advice, and those with unknown discharge status were excluded.

Patients discharged from the hospital with valid data for admission and discharge dates, sex, age, and medical history were included in the study. Those who self-reported smoking at the time of admission were classified as current smokers (hereafter referred to as smokers). No other information on smoking history (duration or intensity) was collected. Former smokers were not identified. Demographic characteristics of the patients are listed in Supporting Information (S1 Table in S1 File).

## Propensity score matching

As shown in Supporting Information (S2 Table in S1 File), demographic characteristics and medical history variables of the 2,239 individuals self-identified as smokers differed significantly from the 29,306 non-smokers, making it difficult to compare smokers with non-smokers. Therefore, we used propensity score matching to limit confounding when examining the association of smoking to study outcome measures. Propensity score was obtained from logistic regression with "nearest" method and "logit" distance where smoking status was the dependent variable and medical history, demographics, medications, and time of hospital admission were independent variables. The final analytic sample was comprised of 6,717 patients with a 1:2 ratio of smokers (2,239) to non-smokers (4,478). As shown in Table 1, there were no statistically significant differences between smokers and non-smokers in age, sex, race, medical history, or medication use after propensity matching, indicating that the two groups were well-matched and balanced.

## Dependent and independent variables and covariates

The primary outcome was severe COVID-19, defined as in-hospital death or the use of mechanical ventilation. The secondary outcome was major adverse cardiac events (MACE), defined as one of these events occurred to patients during hospitalization: acute myocardial infarction, heart failure, cardiogenic shock, ischemic stroke or intracranial hemorrhage, myocarditis, or death by acute myocarditis, heart failure, or stroke. Patients who reported smoking or the use of e-cigarettes (vaping) were categorized as smokers. Covariates included age, sex, race/ethnicity, risk factors and past medical history (see below), medication use, and time-frame of admission. Race/ethnicity was categorized using hierarchical, mutually exclusive categories including Hispanic, non-Hispanic White, non-Hispanic Black, Asian/ Pacific Islander, and Others. Risk factors included obesity (defined as body mass index [BMI] $\geq$30 kg/m$^2$), diabetes mellitus, hypertension, and dyslipidemia. Diabetes mellitus and dyslipidemia were defined as a reported history or use of medications to control these metabolic risk factors. Since data on BMI for nearly 9% of the patients were missing, we imputed an obesity variable applying multiple imputation by chained equations (MICE) using logistic regression models. Past medical history included venous thromboembolism (VTE), coronary artery disease, peripheral artery disease, stroke, heart failure, and chronic kidney disease. Medications prior to admissions that were considered included antiplatelet therapy and anticoagulants. Time-frame of admission for COVID-19 treatment was broken down into five quarters from first to fourth quarter of 2020 and first quarter of 2021 to account for changes in COVID-19 management over time.

## Statistical analysis

Percentages were calculated for categorical variables and compared using Pearson's $\chi^2$- test with Yates' continuity correction. Means and standard deviations were calculated for continuous variables and compared using Student's *t*-test. We generated two multivariable logistic regression models to calculate odds ratios (ORs) to estimate the likelihood of death and use of mechanical ventilation. Odds ratios were estimated by the probabilistic framework of

**Table 1. Univariate analysis of the propensity-matched study population in the AHA COVID-19 CVD Registry from December 2020 to March 2021 stratified by smoking status.**

| | Overall | Non-Smokers | Smokers | Standardized Mean Difference |
|---|---|---|---|---|
| | (N = 6,717) | (N = 4,478) | (N = 2,239) | |
| Age (years) | | | | |
| Mean (SD) | 59.6 (17.8) | 59.4 (18.3) | 60.0 (16.8) | 0.0398 |
| Median [Min, Max] | 61.0 [18.0, 100] | 61.0 [18.0, 100.0] | 62.0 [18.0, 99] | |
| Sex | | | | |
| Male, n (%) | 4,201 (62.5) | 2,818 (62.9) | 1,383 (61.8) | -0.0239 |
| Female, n (%) | 2,516 (37.5) | 1,660 (37.1) | 856 (38.2) | -0.0239 |
| Race/Ethnicity | | | | |
| NH-White, n (%) | 3,465 (51.6) | 2,323 (51.9) | 1,142 (51.0) | -0.0174 |
| Black, n (%) | 1,801 (26.8) | 1,193 (26.6) | 608 (27.2) | 0.0115 |
| Hispanic, n (%) | 922 (13.7) | 615 (13.7) | 307 (13.7) | -0.0006 |
| Asian/Pacific Islanders, n (%) | 191 (2.8) | 128 (2.9) | 63 (2.8) | -0.0027 |
| Other, n (%) | 338 (5.0) | 219 (4.9) | 119 (5.3) | 0.0189 |
| Medical History | | | | |
| Obesity, n (%) | 2,887 (43.0) | 1,934 (43.2) | 953 (42.6) | -0.0126 |
| Diabetes mellitus, n (%) | 2,342 (34.9) | 1,543 (34.5) | 799 (35.7) | 0.0123 |
| Hypertension, n (%) | 4,494 (66.9) | 2,964 (66.2) | 1,530 (68.3) | 0.0461 |
| Dyslipidemia, n (%) | 3,331 (49.6) | 2,189 (48.9) | 1,142 (51.0) | 0.0424 |
| Deep venous/pulmonary embolus, n (%) | 413 (6.1) | 268 (6.0) | 145 (6.5) | 0.0200 |
| Coronary artery disease, n (%) | 1,025 (15.3) | 670 (15.0) | 355 (15.9) | 0.0245 |
| Peripheral artery disease, n (%) | 313 (4.7) | 203 (4.5) | 110 (4.9) | 0.0176 |
| Stroke, n (%) | 716 (10.7) | 457 (10.2) | 259 (11.6) | 0.0426 |
| Heart Failure, n (%) | 1,049 (15.6) | 677 (15.1) | 372 (16.6) | 0.0402 |
| Chronic kidney disease, n (%) | 992 (14.8) | 641 (14.3) | 351 (15.7) | 0.0375 |
| Medications Use | | | | |
| Anti-platelet therapy, n (%) | 2,190 (32.6) | 1,440 (32.2) | 750 (33.5) | 0.0284 |
| Anti-coagulant, n (%) | 810 (12.1) | 524 (11.7) | 286 (12.8) | 0.0321 |
| Time of Admission | | | | |
| First quarter, 2020, n (%) | 1,142 (17.0) | 767 (17.1) | 375 (16.7) | -0.0102 |
| Second quarter, 2020, n (%) | 2,240 (33.3) | 1,496 (33.4) | 744 (33.2) | -0.0038 |
| Third quarter, 2020, n (%) | 1,099 (16.4) | 723 (16.1) | 376 (16.8) | 0.0173 |
| Fourth quarter, 2020, n (%) | 1,980 (29.5) | 1,323 (29.5) | 657 (29.3) | -0.0044 |
| First quarter, 2021, n (%) | 252 (3.8) | 166 (3.7) | 86 (3.8) | 0.0070 |

NH: Non-Hispanic; SD: Standard deviation; AHA: American Heart Association; CVD: Cardiovascular disease

maximum likelihood estimation. Models were adjusted for the indicated demographics, risk factors, medications, and time of admission. We also examined the interaction of smoking with diabetes mellitus, hypertension, race/ethnicity, sex, and age. Statistical significance was assessed at $\alpha = 0.05$. Data analysis was performed using the open-source software R (R Foundation for Statistical Computing, Vienna, Austria).

## Results

### Univariable analysis

The demographic characteristics and medical history variables of the patients who met the inclusion criteria of the study (n = 31,545) are shown in S1 Table in S1 File. After 1:2

**Table 2. Characteristics of the propensity-matched study population of the AHA COVID-19 CVD Registry from December 2020 to March 2021 by survival status.**

| | Overall | Survivors | Death |
|---|---|---|---|
| | (N = 6,717) | (N = 5,825) | (N = 892) |
| **Smoking Status** | | | |
| Smokers, n (%) | 2,239 (33.3) | 1,883 (32.3) | 356 (39.9)* |
| **Age (years)** | | | |
| Mean (SD) | 59.6 (17.8) | 58.1 (17.8) | 69.6 (14.3)* |
| Median [Min, Max] | 61.0 [18.0, 100] | 60.0 [18.0, 100] | 71.0 [18.0, 98.0] |
| **Sex** | | | |
| Male, n (%) | 4,201 (62.5) | 3,602 (61.8) | 599 (67.2)* |
| Female, n (%) | 2,516 (37.5) | 2,223 (38.2) | 293 (32.8)* |
| **Race/Ethnicity** | | | |
| NH-White, n (%) | 3,465 (51.6) | 3,006 (51.6) | 459 (51.5) |
| Black, n (%) | 1,801 (26.8) | 1,559 (26.8) | 242 (27.1) |
| Hispanic, n (%) | 922 (13.7) | 824 (14.1) | 98 (11.0)* |
| Asian/Pacific Islanders, n (%) | 191 (2.8) | 165 (2.8) | 26 (2.9) |
| Other, n (%) | 338 (5.0) | 271 (4.7) | 67 (7.5)* |
| **Medical History** | | | |
| Obesity, n (%) | 2,887 (43.0) | 1,934 (43.2) | 953 (42.6)* |
| Diabetes mellitus, n (%) | 2,342 (34.9) | 1,936 (33.2) | 406 (45.5)* |
| Hypertension, n (%) | 4,494 (66.9) | 3,767 (64.7) | 727 (81.5)* |
| Dyslipidemia, n (%) | 3,331 (49.6) | 2,765 (47.5) | 566 (63.5)* |
| Deep venous/pulmonary embolus, n (%) | 413 (6.1) | 339 (5.8) | 74 (8.3)* |
| Coronary artery disease, n (%) | 1,025 (15.3) | 811 (13.9) | 214 (24.0)* |
| Peripheral artery disease, n (%) | 313 (4.7) | 241 (4.1) | 72 (8.1)* |
| Cerebrovascular disease, n (%) | 716 (10.7) | 578 (9.9) | 138 (15.5)* |
| Heart failure, n (%) | 1,049 (15.6) | 829 (14.2) | 220 (24.7)* |
| Chronic kidney disease, n (%) | 992 (14.8) | 758 (13.0) | 234 (26.2)* |
| **Medications Use** | | | |
| Anti-platelet therapy, n (%) | 2,190 (32.6) | 1,786 (30.7) | 404 (45.3)* |
| Anti-coagulant, n (%) | 810 (12.1) | 638 (11.0) | 172 (19.3)* |
| **Time of Admission** | | | |
| First quarter, 2020, n (%) | 1,142 (17.0) | 901 (15.5) | 241 (27.0)* |
| Second quarter, 2020, n (%) | 2,240 (33.3) | 1,943 (33.4) | 297 (33.3) |
| Third quarter, 2020, n (%) | 1,099 (16.4) | 981 (16.8) | 118 (13.2)* |
| Fourth quarter, 2020, n (%) | 1,980 (29.5) | 1,772 (30.4) | 208 (23.2)* |
| First quarter, 2021, n (%) | 252 (3.8) | 224 (3.8) | 28 (3.1) |

NH: Non-Hispanic; SD: Standard deviation; AHA: American Heart Association; CVD: Cardiovascular disease

* P<0.05 vs survivors

propensity matching on age, sex, race, medical history, and time of hospital admission the final propensity-matched cohort included 6,717 patients (Table 1). Characteristics of the matched sample stratified by mortality status, are listed in Table 2. In this comparison, the percent of smokers in the survivor group was significantly lower (32%) than in the group that died (40%). Those who died were also older than those who survived (70 ± 14 years vs 58 ± 18 years, p<0.05). In addition, those who died were more likely to be male (67% vs 62%, p<0.05). Survival also varied by race. The percent of individuals who were self-reported Non-Hispanic Whites and Asians/Pacific Islanders was higher among those who died, while there were

higher numbers of Hispanics in the survivor group. As reported previously, several characteristics were related to adverse outcomes even in the matched cohort. Those who died also were more likely to have a history of diabetes mellitus, hypertension, dyslipidemia, VTE, coronary artery disease, peripheral artery disease, cerebrovascular disease, and chronic kidney disease. Surprisingly obesity was less prevalent among those who died. The proportions of patients who had used either antiplatelet therapy or anticoagulant medications were significantly higher among non-survivors. In comparison with the first quarter of 2020, patients admitted later during 2020 were less likely to die.

A similar risk profile was seen when the population was stratified by those not receiving mechanical ventilation (n = 5,535) and those receiving respiratory assistance (n = 1,182). Those receiving mechanical ventilation were more likely to be smokers, men, Non-Hispanic Whites, obese, and with a history of diabetes mellitus, hypertension, dyslipidemia, heart failure, chronic kidney disease, and using anti-platelet therapy at the time of admission. Moreover, in comparison with those admitted in the first quarter, those in the fourth quarter were less likely to receive mechanical ventilation (S2 Table in S1 File).

## Multivariable analysis

To estimate the odds of severe outcomes among smokers and non-smokers we examined both adjusted and non-adjusted relationships between smoking status and death or mechanical ventilation in the propensity matched study cohort. As shown in Table 3, smokers had a significantly higher odds of death or mechanical ventilator use (OR, 1.39; 95% CI, 1.20–1.61; and OR 1.16; 95% CI 1.01–1.32, respectively). After adjusting for sociodemographic factors, medical history, medications, and time of admission, smokers had higher adjusted odds of death (adjusted odds ratio [aOR], 1.41; 95% CI, 1.21–1.64) or mechanical ventilator use (aOR, 1.15; 95% CI, 1.01–1.32).

Subgroup analysis indicated that smoking was a stronger risk factor for death in patients between the age of 18–59 years than those more than 60 years of age (Fig 1). Females and males had overlapping risks, although smoking conferred a slightly higher risk in females. Smoking also was associated with higher risks (ORs) among those who were White, obese, with diabetes mellitus, hypertension, chronic kidney disease, who received anticoagulant therapy before hospital admission, or who were admitted second quarter of 2020. Similarly, smoking was associated with elevated risk of mechanical ventilator use in patients who were female, Hispanic, or admitted in the first quarter of 2020 (Fig 2). The point estimate of the association between smoking and death or mechanical ventilator use varied across time intervals of the COVID-19 pandemic, but confidence intervals of these point estimates overlapped considerably.

In analysis of the secondary outcome, we also examined the association between smoking and MACE. Overall, smokers had a significantly higher odds of MACE (OR, 1.29; 95% CI 1.01–1.32). In adjusted analyses, smoking was associated with an increased risk of MACE (aOR 1.27; 95% CI 1.05–1.52). As shown in Fig 3, smoking was associated with increased odds of MACE specifically in those who were less than 60 years of age, female, White, or obese. Smoking was also associated with increased odds of MACE among those admitted during the first quarter of 2020.

## Discussion

The major finding of this study is that in a well-characterized national registry from many different hospitals across the U.S., COVID-19 patients who were identified as current smokers were more likely to die or receive mechanical ventilation than those who were identified as

**Table 3. Multivariate analysis of associations between characteristics and outcomes among the propensity-matched study population of the AHA COVID-19 CVD Registry from December 2020 to March 2021.**

| Exposure, demographics, and risk factors | | Death* | Mechanical ventilator use† |
|---|---|---|---|
| **Smoking status[1]** | | | |
| | Unadjusted | 1.39 (1.20–1.61) | 1.16 (1.01–1.32) |
| | Adjusted[2] | 1.41 (1.21–1.64) | 1.15 (1.01–1.32) |
| **Age, 5 years** | | **1.23 (1.19–1.27)** | 1.01 (0.99–1.04) |
| **Sex** | | | |
| | Male | 1.0 (Reference) | 1.0 (Reference) |
| | Female | **0.80 (0.68–0.93)** | **0.68 (0.59–0.78)** |
| **Race/ethnicity** | | | |
| | White | 1.0 (Ref) | 1.0 (Ref) |
| | Black | 1.12 (0.93–1.35) | 1.17 (0.99–1.37) |
| | Hispanic | 1.04 (0.80–1.34) | 1.17 (0.95–1.43) |
| | NH- Asian/Pacific Islander | 1.09 (0.68–1.69) | 1.05 (0.70–1.54) |
| | NH- Other | **1.81 (1.33–2.46)** | **1.74 (1.33–2.27)** |
| **Medical history** | | | |
| | Obesity | 1.07 (0.91–1.26) | **1.36 (1.18–1.56)** |
| | Diabetes mellitus | **1.26 (1.07–1.48)** | **1.31 (1.13–1.52)** |
| | Hypertension | 1.06 (0.86–1.32) | **1.30 (1.08–1.56)** |
| | Dyslipidemia | 0.88 (0.73–1.07) | 1.01 (0.86–1.19) |
| | Deep vein thrombosis/ | | |
| | Pulmonary embolism | 1.05 (0.79–1.40) | 0.86 (0.65–1.14) |
| | Coronary artery disease | 1.09 (0.89–1.33) | 0.97 (0.79–1.18) |
| | Peripheral artery disease | 1.08 (0.80–1.44) | 0.99 (0.72–1.33) |
| | Cerebrovascular disease | 1.08 (0.87–1.34) | 0.89 (0.71–1.10) |
| | Heart failure | 1.08 (0.89–1.31) | 0.95 (0.79–1.16) |
| | Chronic kidney disease | **1.56 (1.29–1.89)** | 1.09 (0.90–1.31) |
| **Medications** | | | |
| | Anti-platelet | 1.15 (0.96–1.36) | 1.01 (0.86–1.18) |
| | Anti-coagulant | **1.40 (1.13–1.72)** | 1.23 (1.00–1.51) |
| **Time of admission** | | | |
| | First quarter, 2020 | 1.0 (Ref) | 1.0 (Ref) |
| | Second quarter, 2000 | **0.52 (0.43–0.64)** | **0.56 (0.47–0.67)** |
| | Third quarter, 2020 | **0.45 (0.35–0.58)** | **0.51 (0.41–0.62)** |
| | Fourth quarter, 2020 | **0.40 (0.32–0.49)** | **0.35 (0.28–0.42)** |
| | First quarter, 2021 | **0.42 (0.27–0.65)** | **0.38 (0.25–0.57)** |

[1]Smoking status is defined as smoking or e-cigarette (vaping) use. The OR ratios are from a comparison between smokers with a matched group of non-smokers.

[2]Multivariate models were adjusted for age, sex, race/ethnicity, risk factors, medical history, medication use, and the time of hospital admission

NH, non-Hispanic; Obesity is defined as BMI $> = 30$ kg/m$^2$; Bolded OR ratio are statistically significant (P<0.05). Cerebrovascular disease includes stroke and transient ischemic attack (TIA).

* Patient's disposition status is "Expired" at the time of discharge.

† During the hospitalization, intubated or placed on mechanical ventilation.

The study population was matched on medical history, demographics, medications, and time of medications

non-smokers. These analyses provide the most extensive and robust evidence to date that smokers have a higher risk of developing severe COVID-19 and dying as a result of SARS-CoV-2 infection. The relationship between smoking and more severe outcomes was significant

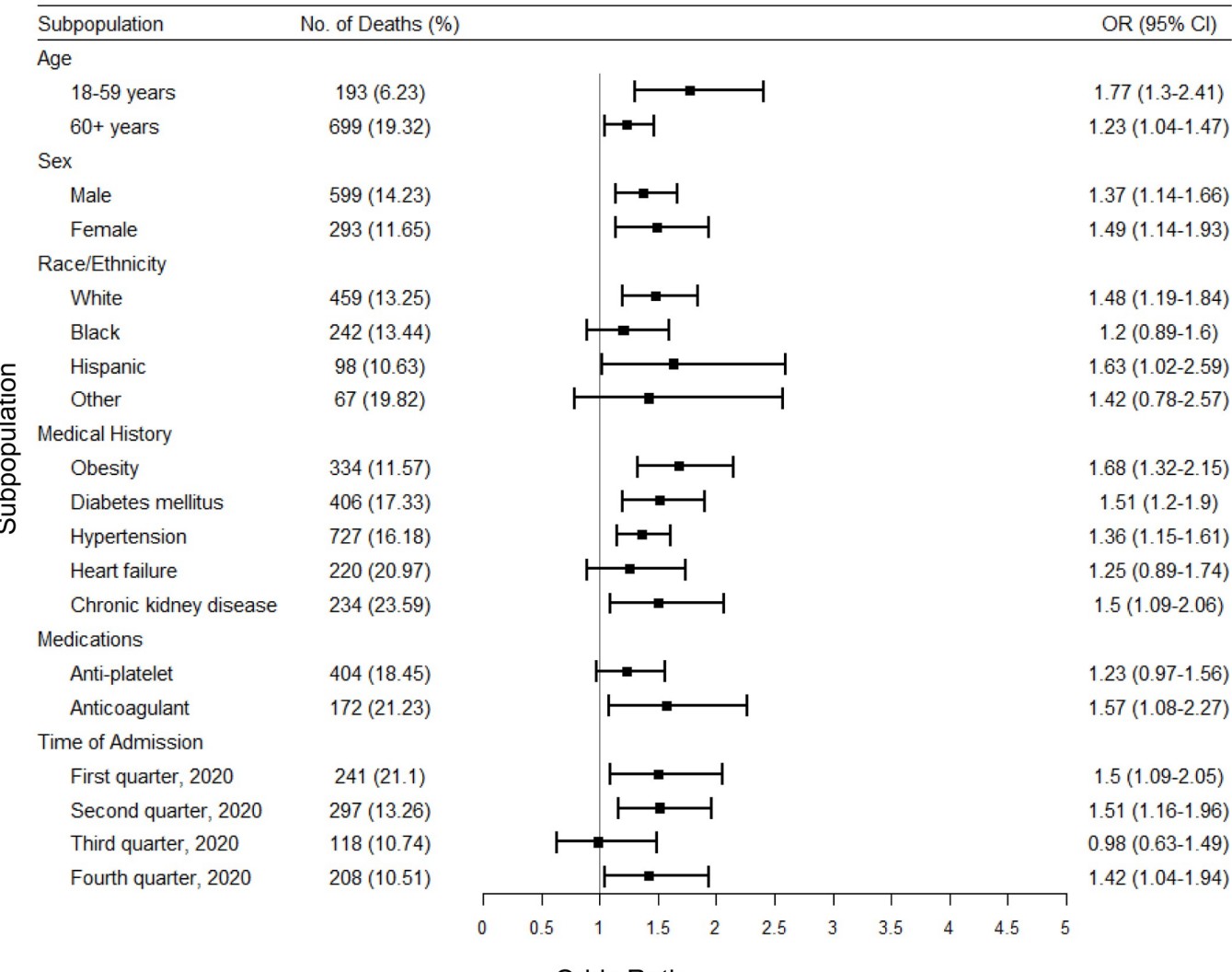

| Subpopulation | No. of Deaths (%) | | OR (95% CI) |
|---|---|---|---|
| **Age** | | | |
| 18–59 years | 193 (6.23) | | 1.77 (1.3-2.41) |
| 60+ years | 699 (19.32) | | 1.23 (1.04-1.47) |
| **Sex** | | | |
| Male | 599 (14.23) | | 1.37 (1.14-1.66) |
| Female | 293 (11.65) | | 1.49 (1.14-1.93) |
| **Race/Ethnicity** | | | |
| White | 459 (13.25) | | 1.48 (1.19-1.84) |
| Black | 242 (13.44) | | 1.2 (0.89-1.6) |
| Hispanic | 98 (10.63) | | 1.63 (1.02-2.59) |
| Other | 67 (19.82) | | 1.42 (0.78-2.57) |
| **Medical History** | | | |
| Obesity | 334 (11.57) | | 1.68 (1.32-2.15) |
| Diabetes mellitus | 406 (17.33) | | 1.51 (1.2-1.9) |
| Hypertension | 727 (16.18) | | 1.36 (1.15-1.61) |
| Heart failure | 220 (20.97) | | 1.25 (0.89-1.74) |
| Chronic kidney disease | 234 (23.59) | | 1.5 (1.09-2.06) |
| **Medications** | | | |
| Anti-platelet | 404 (18.45) | | 1.23 (0.97-1.56) |
| Anticoagulant | 172 (21.23) | | 1.57 (1.08-2.27) |
| **Time of Admission** | | | |
| First quarter, 2020 | 241 (21.1) | | 1.5 (1.09-2.05) |
| Second quarter, 2020 | 297 (13.26) | | 1.51 (1.16-1.96) |
| Third quarter, 2020 | 118 (10.74) | | 0.98 (0.63-1.49) |
| Fourth quarter, 2020 | 208 (10.51) | | 1.42 (1.04-1.94) |

**Fig 1. Multivariate analysis of associations between smoking and death in subpopulations among the propensity-matched study population of the AHA COVID-19 CVD Registry from December 2020 to March 2021.** Death is defined as patient's disposition status "Expired" at the time of discharge. Obesity is defined as BMI $>= 30$ kg/m$^2$. OR, Odds ratio; CI, Confidence interval.

even when the population of smokers was compared with a population of non-smokers with a similar distribution of age, sex, race, and medical history. Moreover, the relationship remained significant after adjusting for demographic and medical history variables, indicating that smoking was associated with more severe COVID-19, independent of age, sex, race, and medical history. Nevertheless, in our subgroup analysis, smoking had a greater impact on outcomes amount younger patients, perhaps because the baseline risk of death in the population is low or because in comparison with the older age group, younger individuals have fewer co-morbidities, which makes smoking a much more significant risk factor. We also found that smokers who were female, White, or were obese, diabetic, or who had chronic kidney disease were more likely to die, which may be indicative of an additive effect of smoking on existing vulnerabilities or comorbidities. Overall, our findings support the notion that smoking is a risk factor for severe outcomes among COVID-19 patients.

Several previous studies have examined the role of pre-existing conditions on susceptibility to SARS-CoV-2 infections and the risk of developing severe COVID-19. In addition to age

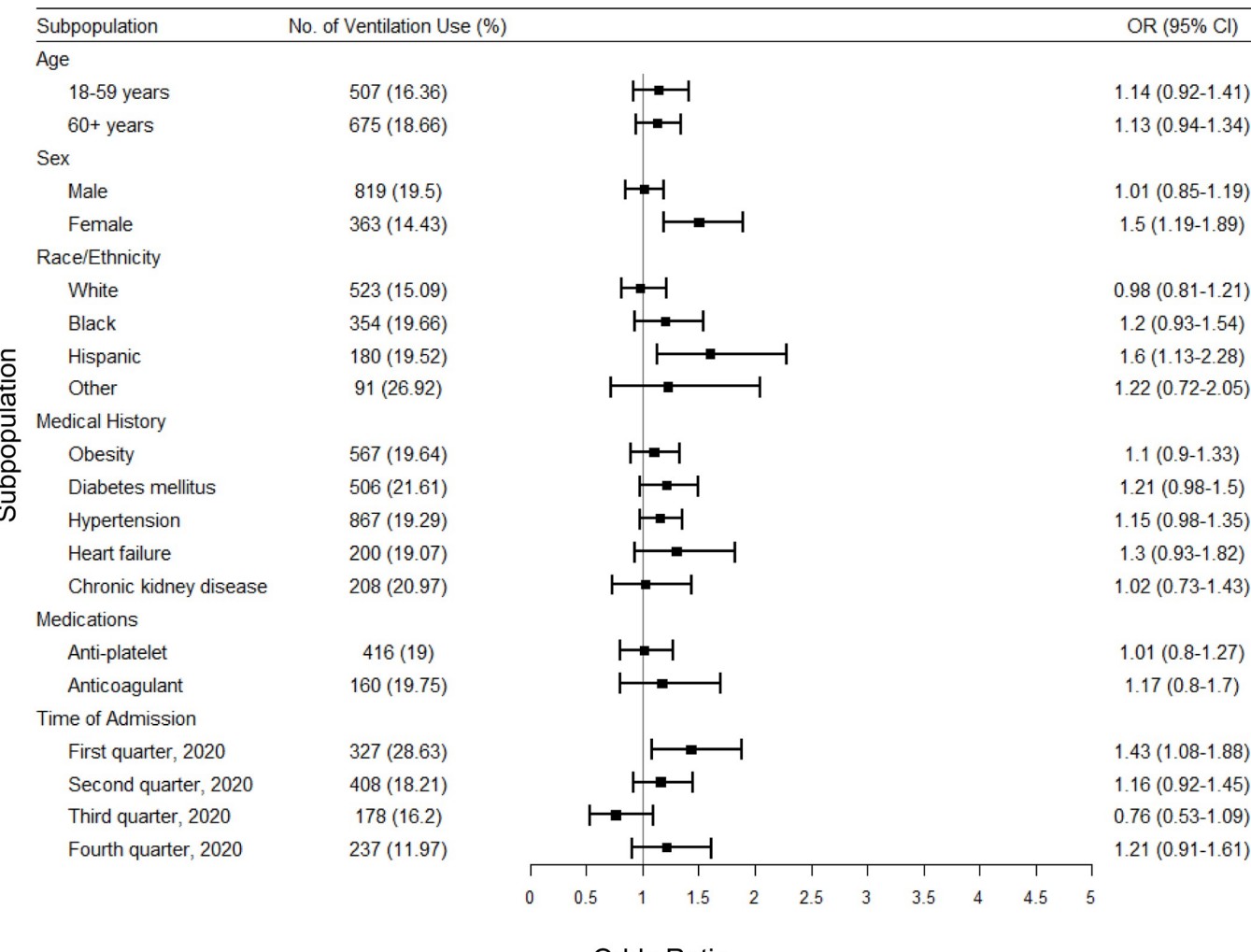

**Fig 2. Multivariate analysis of associations between smoking and mechanical ventilator use in subpopulations among the propensity-matched study population of the AHA COVID-19 CVD Registry from December 2020 to March 2021.** Mechanical ventilation use is defined as the hospitalization, intubated, or placed on mechanical ventilation. Obesity is defined as BMI $>= 30$ kg/m$^2$. OR, Odds ratio; CI, Confidence interval.

and obesity, smoking was reported to be a predictor of mortality in studies from Northern Italy (OR = 2.7, SE = 0.46) [10], and the U.S. (RR = 2.25, CI = 1.39–3.10), and this association was independent of other risk factors [11]. Early data from China identified current smoking as a risk factor for disease progression (OR = 2.51, CI = 1.39–3.32) [9]. However, no association of smoking with COVID-19 was reported in a study from the University Hospital in Padova [12] and subsequent reports from Italy [12] and New York City [13] found that current smoking rates in COVID-19 patients were below those of their respective general populations. In a meta-analysis of data from China, an unusually low prevalence of current smoking was observed, which was approximately one-fourth of the population smoking prevalence [17]. In a risk factor analysis from Oxford, active smoking was linked to decreased odds of a positive SARS-CoV-2 test results [18]. A low prevalence of current smokers among COVID-19 cases (1.3%) compared with the population smoking prevalence in the U.S. (16%) has also been reported by the CDC [19]. Some of these early results may be due to misclassification due to failure to capture a complete and/or accurate smoking history during hospital admissions in the early and hectic days of the pandemic.

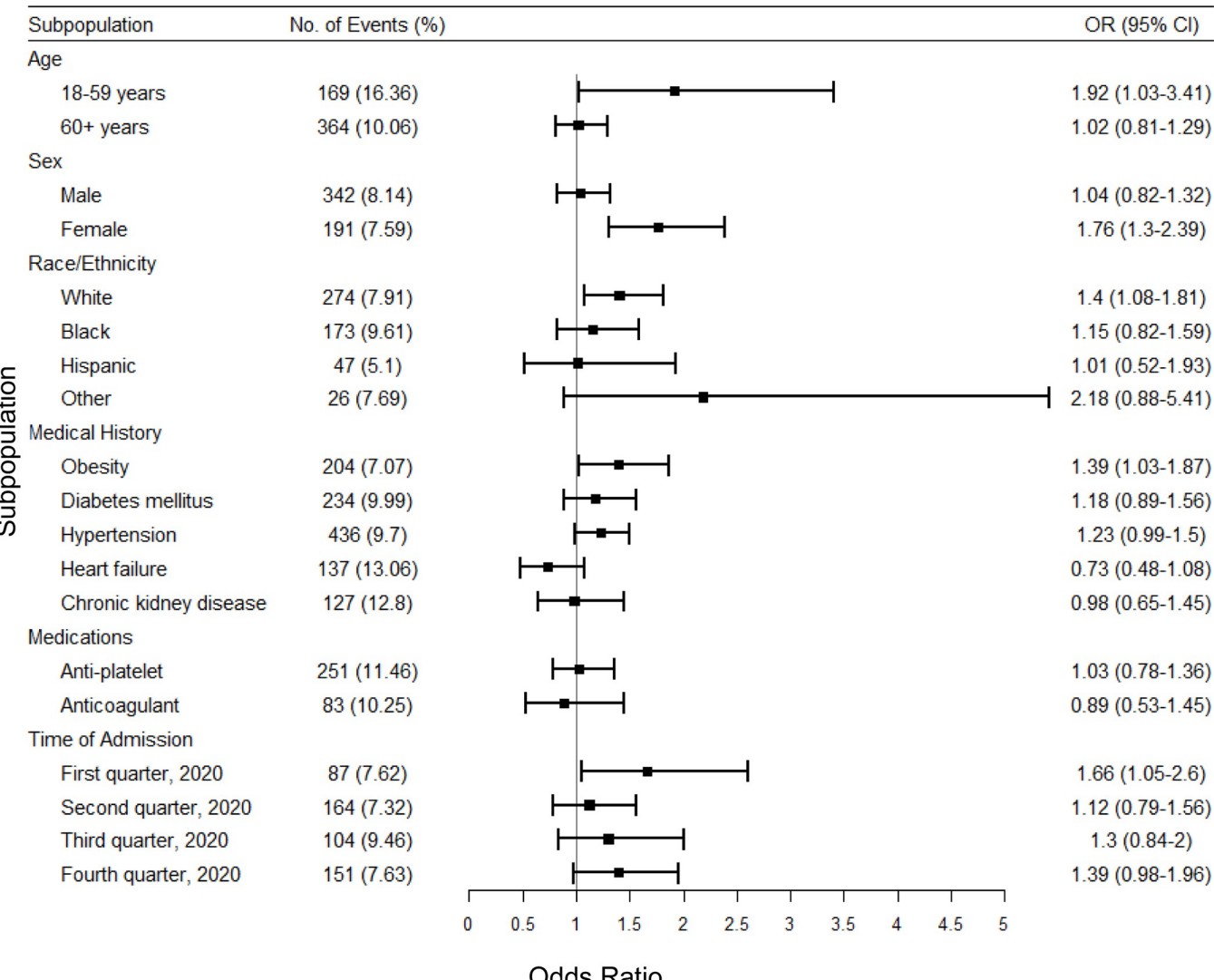

**Fig 3. Multivariate analysis of associations between smoking and major adverse cardiac events (MACE) in subpopulations among the propensity-matched study population of the AHA COVID-19 CVD Registry from December 2020 to March 2021.** Major adverse cardiac events (MACE) is defined as the hospitalization, intubated, or placed on mechanical ventilation. Obesity is defined as BMI > = 30 kg/m². OR, Odds ratio; CI, Confidence interval.

The reduced risk in smokers in some studies is in contrast with a report from England, which found that current smokers and long-term ex-smokers (but not those using nicotine replacement therapy or e-cigarettes) have higher odds of self-reported COVID-19 compared with never smokers [20]. However, in a recent meta-analysis of 32 studies, in comparison with never smokers, current smokers appeared to be at a reduced risk of SARS-CoV-2 infection (RR = 0.74, CI = 0.58–0.93) [21]. A similar pooled meta-analysis of data from over 6,500 patients reported a low prevalence of current smoking among hospitalized patients with COVID-19 [22]. Likewise, data from 7 Italian non-intensive care wards showed an unexpected low (4%) prevalence of current smokers among COVID-19 patients compared to patients admitted for non-COVID-19 disease (16%). It was reported that current smokers were significantly less likely to be hospitalized for COVID-19 compared with non-smokers, even after adjusting for age and gender (OR = 0.14, CI = 0.06–0.31). Hence, the contribution of smoking to risk of SARS-CoV-2 infection remains unclear, and further systematic work is required to elucidate the differential risk of infection among smokers.

In addition to infection susceptibility, smoking has also been reported to be independently associated with hospitalization for COVID-19 [23]. In a meta-analysis of 22 studies, smoking was found to increase the risk of severe disease in hospitalized COVID-19 patients [24]. In a similar meta-analysis of 10 studies, mortality among smokers was 29% compared with 17% among non-smokers (RR = 2.07, CI = 1.59–2.69) [25]. Another meta-analysis reported that both a history of smoking and current smoking were associated with severe COVID-19 cases (OR = 1.51, 95% CI = 1.12–2.05) [26]. However, in contrast to these reports, in an analysis of 10,131 veterans, mortality was associated with older age, male sex and comorbidities, but not smoking [27]. Likewise in a study of 4,353 individuals from Israel, smoking did not significantly increase the risk of COVID-19 complications [28]. Many methodological differences can account for the disparate results among studies, particularly those relating to the sample population, the selection of the comparator group, the diversity of outcomes and the population examined. In this regard, our analysis of data from a wide range of hospitals across the U.S., comparing only hospitalized patients, and following only "hard" outcomes (death, ventilator use, MACE) provides clear and unambiguous evidence that the risk of severe outcomes is higher in COVID-19 patients who smoke when compared with a closely-matched group of non-smokers.

Although the adverse health effects of smoking are well known, the results of this study further reinforce the view that smoking creates a susceptibility state that increases the risk of severe adverse outcomes after SARS-COV-2 infection. Cigarette smoke damages the epithelial barrier which results in increased permeability to inhaled pathogens. It also disrupts the epithelial barrier decreasing mucociliary clearance, leading to the accumulation of inflammatory mucous exudates in small airway lumen [29]. On the other hand, smoking suppresses innate immune response; and nicotine, by binding to the α7nACh receptor, could exert an anti-inflammatory effect by inhibiting NF-κB activation [30]. How these opposing effects of nicotine or smoking affect susceptibility to SARS-CoV-2 infection, immune responses to the virus or progression to severe disease remains unclear. Nonetheless, the robust and significant increase in the risk of severe COVID-19 seen in our study, particularly among young individuals, underscores the urgent need for extensive public health interventions such as anti-smoking campaigns and increased access to cessation therapy, especially in the age of COVID.

## Limitations

Although our study has many strengths, it has significant limitations. Complete smoking history was not available, so we could not distinguish between never smokers and former smokers. Moreover, smoking status was identified by self-report and could not be independently verified, and we had no information on duration (pack years) and intensity (cigarettes smoked per day) of smoking. However, such exposure misclassification is likely to diminish the effect size as such differences regress to the mean. In addition, we have limited data on biomarkers of inflammation or coagulation so we could not assess whether smokers had higher rates of inflammation or thrombosis. Because we only examined those admitted to the hospital, we could not assess how smoking affects susceptibility to SARS-CoV-2 infection. Finally, although we utilized propensity matching and multivariable logistic modeling to account for a wide variety of variables that are potentially associated with smoking and/or the outcome of death or mechanical ventilation, residual confounding is always possible in observational studies.

## Conclusion

Among a large population of patients admitted for COVID-19, smoking was associated with a higher risk of severe COVID-19, including death or mechanical ventilation, independent of sociodemographic characteristics and medical history.

## Supporting information

**S1 File.**
(DOCX)

## Author Contributions

**Conceptualization:** Lori B. Daniels, Andrew P. DeFilippis, Naomi M. Hamburg, Rachel J. Keith, Revanthy Sampath Kumar, Andrew C. Strokes, Rose Marie Robertson, Aruni Bhatnagar.

**Data curation:** Yosef Khan.

**Formal analysis:** Ram Poudel, Andrew P. DeFilippis, Naomi M. Hamburg, Yosef Khan, Andrew C. Strokes, Aruni Bhatnagar.

**Funding acquisition:** Rose Marie Robertson, Aruni Bhatnagar.

**Methodology:** Ram Poudel, Andrew P. DeFilippis.

**Software:** Ram Poudel.

**Supervision:** Aruni Bhatnagar.

**Writing – original draft:** Lori B. Daniels, Andrew P. DeFilippis, Naomi M. Hamburg, Rachel J. Keith, Revanthy Sampath Kumar, Andrew C. Strokes, Rose Marie Robertson, Aruni Bhatnagar.

**Writing – review & editing:** Lori B. Daniels, Andrew P. DeFilippis, Naomi M. Hamburg, Rachel J. Keith, Revanthy Sampath Kumar, Andrew C. Strokes, Rose Marie Robertson, Aruni Bhatnagar.

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
