## [Decision Letter · Decision Letter 0]

25 Jan 2022

PONE-D-22-00192Smoking is Associated with Increased Risk of Cardiovascular Events, Disease Severity, and Mortality Among Patients Hospitalized for SARS-CoV-2 InfectionsPLOS ONE

Dear Dr. Bhatnagar,

Thank you for submitting your manuscript to PLOS ONE. After careful consideration, we feel that it has merit but does not fully meet PLOS ONE’s publication criteria as it currently stands. Therefore, we invite you to submit a revised version of the manuscript that addresses the points raised during the review process.

We look forward to receiving your revised manuscript.

Kind regards,

**Gaetano Santulli, MD**

Academic Editor

PLOS ONE

Journal Requirements:

2. In ethics statement in the manuscript and in the online submission form, please provide additional information about the patient records/samples used in your retrospective study. Specifically, please ensure that you have discussed whether all data/samples were fully anonymized before you accessed them and/or whether the IRB or ethics committee waived the requirement for informed consent. If patients provided informed written consent to have data/samples from their medical records used in research, please include this information

3. You indicated that ethical approval was not necessary for your study. We understand that the framework for ethical oversight requirements for studies of this type may differ depending on the setting and we would appreciate some further clarification regarding your research. Could you please provide further details on why your study is exempt from the need for approval and confirmation from your institutional review board or research ethics committee (e.g., in the form of a letter or email correspondence) that ethics review was not necessary for this study? Please include a copy of the correspondence as an ""Other"" file.

The strengths and limitations of the study should be deeply addressed, taking into account sources of potential bias or imprecision: Discuss both direction and magnitude of any potential bias.

The fundamental role of endothelial dysfunction in the systemic manifestations of COVID-19 should be better discussed:

-J Clin Med. 2020;9(5):1417; doi: 10.3390/jcm9051417

-Expert Opin Ther Targets. 2020 Jun 27:1-8. doi: 10.1080/14728222.2020.1783243

-Cells. 2020 Jul 9;9(7):E1652. doi: 10.3390/cells9071652

-Eur Heart J. 2021 May 23;7(3):e2-e3. doi: 10.1093/ehjcvp/pvaa145.

-Atherosclerosis. 2021;322:39-50. doi: 10.1016/j.atherosclerosis.2021.02.009.

-EClinicalMedicine. 2021 Sep 9:101125. doi: 10.1016/j.eclinm.2021.101125.

-Oxid Med Cell Longev. 2021 Aug 21;2021:8671713

-Theranostics. 2021;11(16):8076-8091.

-Crit Care. 2021 Aug 25;25(1):306.

-Front Physiol. 2020 Aug 4;11:989. doi:10.3389/fphys.2020.00989 (chronological appraisal of the publications on COVID-19 and endothelial dysfunction).

Reviewers' comments:

Reviewer's Responses to Questions

**Comments to the Author**

1. Is the manuscript technically sound, and do the data support the conclusions?

Reviewer #1: Yes

Reviewer #2: Yes

2. Has the statistical analysis been performed appropriately and rigorously? 

Reviewer #1: Yes

Reviewer #2: Yes

3. Have the authors made all data underlying the findings in their manuscript fully available?

Reviewer #1: Yes

Reviewer #2: No

4. Is the manuscript presented in an intelligible fashion and written in standard English?

Reviewer #1: Yes

Reviewer #2: Yes

5. Review Comments to the Author

Reviewer #1: The paper aimed to investigate the association of smoking and Covid-19 in hospitalized patients. The idea is interesting and the study has been presented in a scientific manner. I have just few concerns:

- I see that data come from previously published AHA registries. Howevere, did you obtain the ethical approval from your committe? If no, please clarify the reason.

- Please improve the discussion and the reference section discussing and citing the following PMID:

34655476

35027590

33540664

32464099

33246296

32341101

Reviewer #2: The data and analysis in this paper are fine, but there are several issues that need to be resolved before the paper can be accepted.

1. The association between smoking and COVID-19 disease severity among people diagnosed with COVID is not "limited" or "unclear." There have been several meta-analyses published on this question (beginning with Patanavanich R, Glantz SA. Smoking Is Associated With COVID-19 Progression: A Meta-analysis. Nicotine Tob Res. 2020 Aug 24;22(9):1653-1656. doi: 10.1093/ntr/ntaa082. PMID: 32399563; PMCID: PMC7239135) which show a positive association using a variety of approaches. Thus, the authors overstate to marginal value of their paper. The actual contribution is to confirm this association in a well-characterized group of patients in the US (which is more than enough to warrant publication in PLOS One). This language throughout the paper needs to be toned down accordingly.

2. This situation is different from the question of how smoking affects risk of COVID infection which has been more equivocal, as illustrated in Ref 14 of the manuscript. Because the present manuscript deals only with hospitalized patients, it does not provide any information about the relationship between smoking can COVID risk. This distinction needs to be made clear throughout the paper. Beyond making this distinction and commenting on the lack of consensus on the effect of smoking on risk of COVID infection, discussion of this question should be dropped from the paper, including the Discussion section (pages 7 and * because the data and analysis in this paper does not contribute anything to the discussion of smoking and risk of COVID infection.

3. The authors lump e-cigarette use (vaping) in with smoking in their analysis. They need to justify this approach explicitly, both in terms of biology and by doing a sensitivity analysis showing that doing so does not affect the results. The fact that smoking and vaping are combined needs to be indicated in the abstract. Indeed, if smoking and vaping have similar increases in risk, that would be an important result on its own.

4. The finding that the risks associated with smoking are higher in younger patients is consistent with a recent meta-analysis that reached this conclusion based on data pooled across studies (Patanavanich R, Glantz SA. Smoking is associated with worse outcomes of COVID-19 particularly among younger adults: a systematic review and meta-analysis. BMC Public Health. 2021 Aug 16;21(1):1554. doi: 10.1186/s12889-021-11579-x. PMID: 34399729; PMCID: PMC8366155). The fact that the authors found this result in a single dataset is an important contribution of this paper.

5. The authors state that the data are available but do not say where it is deposited so that people can access it.

6. PLOS authors have the option to publish the peer review history of their article (what does this mean?). If published, this will include your full peer review and any attached files.

Reviewer #1: No

Reviewer #2: No

---

## [Author Response · Author response to Decision Letter 0]

11 Jun 2022

Response to Reviewers’ Comments

Reviewer # 1: 

1. “I see the data come from previously published AHA registries. However, did you obtain ethical approval from your committee?”

Get-With-The-Guidelines® (GWTG) is a continuous quality improvement program registry, overseen by the American Heart Association (AHA) and an Oversight Committee of clinician leaders in quality improvement, that retrospectively collects patient-level adherence data within 3,000 U.S. hospitals. To participate, hospital sites directly and retrospectively abstract or upload quality of care and outcomes data from hospitalized patients’ electronic health records into the registry. Demographic data, clinical data, and test results obtained as part of routine care for CVD patients, as well as pertinent treatments and in-hospital outcomes, are captured. Patient identities (e.g. name, social security number, medical record number, etc.) are not collected. Sites are responsible for assigning a unique study identifier to each submitted record, which is not shared outside the site so the patient cannot be re-identified. The data are collected for quality improvement purposes, and therefore patients are neither recruited nor consented. All research completed using analyses of the AHA GWTG program has been determined to be exempt from IRB oversight by Advarra (see attached). This has been added to the revised text (see page 3).

2. “Please improve the discussion and the reference section discussing and citing the following PMID”

Thank you for pointing us to the relevant literature. To the extent possible, we have cited these references both in the introduction and the discussion.

Reviewer # 2:

1. “The association between smoking and COVID-19 disease severity among people diagnosed with COVID is not "limited" or "unclear." There have been several meta-analyses published…... This language throughout the paper needs to be toned down accordingly.”

The discussion has been “toned down” as suggested. The “limited” and “unclear” qualifications have been removed, although we do note that not all studies have led to consistent results. 

2. “This situation is different from the question of how smoking affects risk of COVID infection which has been more equivocal, as illustrated in Ref 14 of the manuscript. Because the present manuscript deals only with hospitalized patients, it does not provide any information about the relationship between smoking can COVID risk. This distinction needs to be made clear throughout the paper. Beyond making this distinction and commenting on the lack of consensus on the effect of smoking on risk of COVID infection, discussion of this question should be dropped from the paper, including the Discussion section (pages 7 and * because the data and analysis in this paper does not contribute anything to the discussion of smoking and risk of COVID infection.”

We have removed the discussion of the relationship between smoking and the risk of COVID-19 infection.

3. “The authors lump e-cigarette use (vaping) in with smoking in their analysis. They need to justify this approach explicitly, both in terms of biology and by doing a sensitivity analysis showing that doing so does not affect the results. The fact that smoking and vaping are combined needs to be indicated in the abstract. Indeed, if smoking and vaping have similar increases in risk, that would be an important result on its own.”

The number of individuals in our registry who used e-cigarettes is very small – 41 patients reported e-cig only versus 2198 who reported smoking only without e-cigarettes. In our sensitivity analysis, removal of e-cigarette users from the dataset did not significantly affect the odds ratio of in-hospital death or mechanical ventilator use. This has been stated in the revised manuscript. 

4. “The finding that the risks associated with smoking are higher in younger patients is consistent with a recent meta-analysis that reached this conclusion based on data pooled across studies … The fact that the authors found this result in a single dataset is an important contribution of this paper.”

Thank you for pointing this out. We have cited the paper in our revised manuscript.

5. “The authors state that the data are available but do not say where it is deposited so that people can access it.”

The American Heart Association has a strict policy regarding the use and integrity of its Get-With-The-Guidelines® data and is unable to provide public access to the COVID-19 dataset, in full or in part without other agreements in place. However, researchers are able to go to www.precision.heart.org , navigate to data, documentation, and COVID-19 to see data documentation, dictionary and coding information and explore the variables, descriptions, source, type, missingness and distribution of the data.

---

## [Editor Report · Decision Letter 1]

17 Jun 2022

Smoking is Associated with Increased Risk of Cardiovascular Events, Disease Severity, and Mortality Among Patients Hospitalized for SARS-CoV-2 Infections

PONE-D-22-00192R1

Dear Dr. Bhatnagar,

We’re pleased to inform you that your manuscript has been judged scientifically suitable for publication and will be formally accepted for publication once it meets all outstanding technical requirements.

Kind regards,

Gaetano Santulli, MD

Academic Editor

PLOS ONE

---

## [Editor Report · Acceptance letter]

7 Jul 2022

PONE-D-22-00192R1 

Smoking is Associated with Increased Risk of Cardiovascular Events, Disease Severity, and Mortality Among Patients Hospitalized for SARS-CoV-2 Infections 

Dear Dr. Bhatnagar:

I'm pleased to inform you that your manuscript has been deemed suitable for publication in PLOS ONE. Congratulations! Your manuscript is now with our production department. 

Kind regards, 

on behalf of

Professor Gaetano Santulli 

Academic Editor

PLOS ONE